# Uncovering Biological Motifs and Syntax via Sufficient and Necessary Explanations

## Abstract

In recent years, deep neural networks (DNNs) have excelled at learning from high-throughput genome-profiling experiments to predict transcription factor (TF) binding. TF binding is driven by sequence motifs, and explaining how and why DNNs make accurate predictions could help identify these motifs, as well as their logical syntax. However, the black-box nature of DNNs makes interpretation difficult. Most post-hoc methods evaluate the importance of each base pair in isolation, often resulting in noise since they overlook the fact that motifs are contiguous regions. Additionally, these methods fail to capture the complex interactions between different motifs. To address these challenges, we propose Motif Explainer Models (MEMs), a novel explanation method that uses sufficiency and necessity to identify important motifs and their syntax. MEMs excel at identifying multiple disjoint motifs across DNA sequences, overcoming limitations of existing methods. Moreover, by accurately pinpointing sufficient and necessary motifs, MEMs can reveal the logical syntax that governs genomic regulation.

## 1 Introduction

The regulatory function of DNA sequences is determined by short DNA segments called "motifs", where certain proteins called "transcription factors" (TFs) bind to regulate gene activity (Klug, 1995; Alberts et al., 2002; Siggers & Gordân, 2014; Lambert et al., 2018). TF binding depends on the arrangement and logical combination of these motifs. High-throughput experiments provide a genome-wide view of regulatory activity in various cell types (Consortium et al., 2012). Techniques like DNase-seq and ATAC-seq identify regions of open chromatin where TFs can bind, while more targeted methods like ChIP-seq offer insights into specific protein–DNA interactions.

Given the complexity and volume of data from these experiments, accurately predicting genome activity from DNA sequences requires models that can handle large datasets and capture the intricate interactions and arrangement of motifs. Deep neural networks (DNNs), with their proven success in various biological sequence prediction tasks, are well-suited for this challenge. They have achieved state-of-the-art performance in predicting TF binding from DNA sequences (Alipanahi et al., 2015; Avsec et al., 2021; Eraslan et al., 2019). Genomic DNNs take DNA sequences as inputs and learn to predict a label from a regulatory profiling experiment, such as whether a TF binds to a given sequence. The goal is to use these accurate models to identify the motifs and syntax governing genomic regulation (Novakovsky et al., 2023).

Despite their success, the black-box nature of these models makes it difficult to understand how and why they make specific predictions (Zednik, 2021; Tomsett et al., 2018). In regulatory genomics, this has led to the development of post-hoc methods to explain genomic DNNs at the local (sample) level. Given a model $f$ and an input $N$-length DNA sequence $\mathbf{x} = (x_1, x_2, \ldots, x_N) \in \mathbb{R}^N$, these methods aim to identify important motifs for the prediction $f(\mathbf{x})$ by assigning an importance score to each base pair $x_i$, and then segmenting out high-importance regions as putative motifs. However, many of these methods fall short because they evaluate the importance of individual base pairs in *isolation*, and do not leverage the fact that motifs are short, contiguous regions. Furthermore, these methods also fail to capture the complex interactions between motifs.

To address these challenges, Linder et al. (2022) introduced *scramblers*, a model-based explanation method optimized for the discrete nature of biological sequences. Scramblers outperform other post-hoc methods by identifying important motifs using learned stochastic masks. However, when

the input is a complex DNA sequence with multiple motifs, scramblers struggle to identify multiple short contiguous regions as they do not leverage the key properties of motifs we know of. Due to this shortcoming, like other methods, scramblers also cannot uncover the logical syntax governing regulatory behavior.

In this work, we propose a novel model-based explanation method called *Motif Explainer Models* (MEMs), which leverages the concepts of sufficiency and necessity to produce meaningful explanations. For complex DNA sequences composed of disjoint and contiguous motifs, we show that our method accurately identifies sufficient and necessary motifs, and outperforms the current state-of-the-art method of scramblers. Furthermore, by employing both sufficient and necessary explanations together, MEMs reveal the logical syntax between motifs that governs genomic regulation.

## 1.1 RELATED WORKS

Several methods have been proposed for discovering motifs from genomic DNNs.

**Visualizing Convolutional Filters.** Nearly all genomic DNNs are convolutional in their first layers (Alipanahi et al., 2015; Zhou & Troyanskaya, 2015; Kelley et al., 2016; Avsec et al., 2021). To identify important motifs, many works have focused on visualizing CNN filters (Alipanahi et al., 2015; Kelley et al., 2016) because early convolutional layers often capture basic patterns, while deeper layers capture more complex features (Zeiler & Fergus, 2014; Yosinski et al., 2015; Simonyan et al., 2014). However, this approach has shown limited success, as it assumes each filter learns one motif, and each motif is learned by one filter. Recent research shows that motifs are distributed across multiple filters and layers (Tseng et al., 2024).

**Measuring Influence via Post-hoc Explanation Methods.** Popular post-hoc explanation methods like CAM (Zhou et al., 2016), LIME (Ribeiro et al., 2016), gradient-based approaches (Selvaraju et al., 2017; Shrikumar et al., 2017; Jiang et al., 2021), Shapley value-based methods (Chen et al., 2018; Teneggi et al., 2022), and perturbation-based methods (Fong & Vedaldi, 2017; Fong et al., 2019) have been adapted to identify motifs. These methods assign importance scores to each base pair (Sundararajan et al., 2017; Shrikumar et al., 2017), but they perform poorly for two reasons. First, by evaluating base pairs individually, they miss key motifs because subsequences inherently interact in complex ways to regulate function. Second, these methods are computationally expensive. For instance, integrated gradients and DeepLIFT require integrating over the entire DNN, while Shapley-based methods require exponential computations (Strumbelj & Kononenko, 2010; Lundberg & Lee, 2017). Additionally, the importance scores are often noisy, fragile, and fail to reveal the model's true decision-making process (Ghorbani et al., 2019; Tseng et al., 2020), making it difficult to rely on downstream tools like MoDISco to cluster them into motifs (Shrikumar et al., 2018).

**Scramblers.** To overcome these limitations, Linder et al. (2022) proposed scramblers, a model-based explanation method that learns stochastic masks to highlight the base pairs crucial for predictions. Scramblers predict position-specific scoring matrices (PSSMs), where unimportant base pairs are "scrambled" by increasing their entropy. Scramblers have a distinct advantage over many traditional post-hoc explanation methods due to their model-based approach: a scrambler only needs to be trained once for any model predictive $f$, after which importance scores for any query DNA sequence can be obtained in a single evaluation. While scramblers outperform other post-hoc methods, they still struggle with sequences composed of multiple disjoint and contiguous motifs. This is due to a regularization penalty that focuses on controlling entropy, rather than incorporating the core characteristics of motifs (small, contiguous, and disjoint). As a result, scramblers are limited in complex settings and fail to uncover the logical syntax of motif interactions for genomic regulation.

## 1.2 SUMMARY OF OUR CONTRIBUTIONS

We address the challenges of interpreting genomic DNNs by proposing a novel model-based explanation method focused on identifying important motifs and the logical syntax governing gene regulation. Specifically, our method can identify both sufficient or necessary motifs for predictions, providing more accurate and interpretable explanations for genomic DNNs on complex DNA sequences. Our contributions include:

1. Motif Explainer Models (MEMs): We introduce *Motif Explainer Models* (MEMs), a model-based explanation method capable of generating sufficient or necessary explanations for genomic DNNs. MEMs can handle disjoint and contiguous motifs gracefully, capturing the intricate arrangements and interactions that other methods miss.

2. Uncovering Logical Syntax via Sufficiency and Necessity: By combining sufficient and necessary explanations, we show that MEMs can reveal the logical syntax governing how motifs interact to regulate downstream gene expression.

3. Experimental Validation: Through a series of experiments, we demonstrate that MEMs outperform scramblers, the current state-of-the-art method in identifying important motifs. Additionally, we show how MEMs can deduce common biological syntactical rules, such as cooperation, repression, and redundancy.

## 2 BACKGROUND

**Notation.** Random vectors and their observed values are denoted with boldface uppercase (e.g., $\mathbf{X}$) and lowercase (e.g., $\mathbf{x}$) letters. For a subset of features $S \subseteq [N]$ (where $[N] := \{1, \dots, N\}$), we denote its complement as $\bar{S} = [N] \setminus S$. Additionally, subscripts index features, e.g. the vector $\mathbf{x}_S$ is the restriction of $\mathbf{x}$ to the components indexed by $S$. The input domain of $N$-length DNA sequences and output domain of binary labels are denoted as $\mathcal{X} = \{\text{A}, \text{C}, \text{G}, \text{T}\}^N$ and $\mathcal{Y} = \{0, 1\}$, respectively. A distribution over features and labels $\mathcal{X} \times \mathcal{Y}$ is denoted as $\mathcal{D}$ and for such a distribution, the marginal distribution over features is represented as $\mathcal{D}_{\mathcal{X}}$. Lastly, denote $\rho : \mathbb{R} \times \mathbb{R} \mapsto \mathbb{R}$ to be any symmetric function that measures the similarity between elements $a, b \in \mathbb{R}$ with the property $\rho(a, b) = 0 \iff a = b$. A common choice is $\rho(a, b) = |a - b|$, which we use in our experiments.

**Setting.** We consider a binary classification setting with an unknown distribution $\mathcal{D}$ over $\mathcal{X} \times \mathcal{Y}$, a domain of $N$-length DNA sequences and binary labels. Since the inputs are $N$-length DNA sequences, when we refer to an input DNA sequence $\mathbf{x}$, note it is implicitly being expressed as a one-hot encoded pattern, $\mathbf{x} \in \{0, 1\}^{N \times 4}$ (a $N$-length sequences of alphabet size 4, representing the 4 base pairs $\{\text{A}, \text{C}, \text{G}, \text{T}\}$). We assume access to a differentiable predictor $f : \mathcal{X} \mapsto \mathcal{Y}$, pretrained on DNA sequence–label pairs, $(\mathbf{X}, Y) \sim \mathcal{D}$. Our goal of interpretation is: for a fixed DNA sequence $\mathbf{x}$, identify which short subsequences, i.e. motifs, in $\mathbf{x}$ are most important for the prediction $f(\mathbf{x})$. To do so, our method—as with many other post-hoc methods (Covert et al., 2021; Fong & Vedaldi, 2017; Fong et al., 2019)—relies on evaluating how a predictor's behavior changes when base-pairs in $\mathbf{x}$ are retained or omitted. Since $f$ can only accept $N$-length sequences as an input we employ the standard technique for querying $f$ on subsets of features by evaluating the *average restricted prediction*

$$f_S(\mathbf{x}) = \underset{\mathbf{X}_{\bar{S}} \sim \mathcal{V}_{\bar{S}}}{\mathbb{E}}[f(\mathbf{x}_S, \mathbf{X}_{\bar{S}})] \tag{1}$$

where $\mathbf{x}_S$ is fixed and $\mathbf{X}_{\bar{S}}$ is a random vector sampled from $\mathcal{V}_{\bar{S}}$, the marginal distribution, over $S$, of an arbitrary reference distribution $\mathcal{V}_{[N]}$ (Covert et al., 2021; Teneggi et al., 2023; Bharti et al., 2024).

### 2.1 SUFFICIENCY AND NECESSITY

Our method takes as input a pretrained predictor $f : \mathcal{X} \mapsto \mathcal{Y}$ and a fixed DNA sequence $\mathbf{x}$, and outputs a subset $S \subseteq [d]$ that is considered "important" for the prediction $f(\mathbf{x})$. We define the importance of $S$ using slightly modified notions of sufficiency and necessity originally proposed by Bharti et al. (2024). We present our modified definitions, below for clarity:

**Definition 1** (Sufficiency & Necessity (Bharti et al., 2024)). *Let $\epsilon$ and $\Delta > 0$. Denote $\rho : \mathbb{R} \times \mathbb{R} \mapsto \mathbb{R}$ to be a similarity measure. For a predictor $f$ and sample $\mathbf{x}$, denote $\hat{Y}(\mathbf{x}) = \mathbb{1}[f(\mathbf{x}) \geq 0.5]$ to be the predicted class of $\mathbf{x}$ by $f$.*

*A subset $S \subseteq [d]$ is $\epsilon$-sufficient with respect to a distribution $\mathcal{V}$ for $f$ at $\mathbf{x}$ if*

$$\rho(\hat{Y}(\mathbf{x}), f_S(\mathbf{x})) \leq \epsilon \tag{2}$$

*A subset $S \subseteq [d]$ is $\Delta$-necessary with respect to a distribution $\mathcal{V}$ for $f$ at $\mathbf{x}$ if*

$$\rho(\hat{Y}(\mathbf{x}), f_{\bar{S}}(\mathbf{x})) \geq \Delta. \tag{3}$$

In other words, this definition of sufficiency states that, for a reference distribution $\mathcal{V}$, a subset of features $S$ is $\epsilon$-sufficient if, with $\mathbf{x}_S$ fixed, the average restricted prediction $f_S(\mathbf{x})$ is $\epsilon$ close to the predicted class $\hat{Y}(\mathbf{x})$. Conversely, the definition of necessity states that, for $\mathcal{V}$, a subset $S$ is $\Delta$-necessary if, when the features in $S$ are marginalized out, the resulting average restricted prediction $f_{\bar{S}}(\mathbf{x})$ is $\Delta$ away from the predicted class $\hat{Y}(\mathbf{x})$. Later in this work, we will demonstrate why accurate sufficient *and* necessary explanations are essential for deducing motif syntax. Furthermore, we highlight that while scramblers attempt to generate these explanations with some success, our proposed method achieves far greater accuracy, enabling more reliable deduction of the underlying motifs and their logic.

## 2.2 Scramblers

Given a pre-trained predictor $f$ and fixed DNA-sequence $\mathbf{x}$, a scrambler (Linder et al., 2022) is a learned model $g : \mathcal{X} \mapsto \mathbb{R}_{>0}^N$ that predicts a set of real-valued importance scores in $(0, \infty]^N$. These scores produce a probability distribution $P_g(\mathbf{x})$ that we can sample from. Specifically, $P_g(\mathbf{x})$ is a set of $N$ categorical softmax-nodes, also known as a position-specific scoring matrix (PSSM) that interpolates between $\mathbf{x} \in \{0, 1\}^{N \times 4}$ and a non-informative background distribution $\tilde{B} \in [0, 1]^{N \times 4}$. One can learn a scrambler $S$ by solving the following optimization problem

$$\underset{g \subseteq \mathcal{H}}{\arg\min} \quad \underset{\mathbf{X} \sim \mathcal{D}_{\mathcal{X}}}{\mathbb{E}} \left[ L(f, \mathbf{X}, P_g) + \lambda \cdot \text{Con}(P_g(\mathbf{X}), \tilde{B}, \mathbf{X}) \right] \tag{4}$$

where $L(f, \mathbf{X}, P_g)$ denotes a loss function, $\text{Con}(P_g(\mathbf{X}), \tilde{B}, \mathbf{X})$ a conservation penalty, and $\lambda > 0$ a hyperparameter which controls the magnitude of the penalty. Depending on the type of scrambler one wants to learn, the loss function $L(f, \mathbf{X}, P_g)$, the conservation penalty, and the functional form of probability distribution $P_g(\mathbf{X})$ will vary. There are two types of scramblers, an inclusion scrambler and occlusion scrambler.

**Inclusion Scrambler.** An *inclusion* scrambler is trained with

$$L(f, \mathbf{X}, P_g) = \underset{\tilde{\mathbf{X}} \sim P_g(\mathbf{X})}{\mathbb{E}} \left[ \text{KL} \left[ f(\tilde{\mathbf{X}}) \| f(\mathbf{X}) \right] \right] \tag{5}$$

$$\text{Con}(P_g(\mathbf{X}), \tilde{B}, \mathbf{X}) = \left( t_{\text{bits}} - \frac{1}{N} \text{KL} \left[ \tilde{B} \| P_g(\mathbf{X}) \right] \right)^2 \tag{6}$$

$$P_g(\mathbf{x}) = \sigma(\log(\tilde{B}) + \mathbf{x} \times \dot{g}(\mathbf{x})). \tag{7}$$

where $\sigma$ denotes the softmax $\sigma(L)_{ij} = \frac{\exp(L_{ij})}{\sum_{k=1}^M \exp(L_{ik})}$ and $\dot{g}(\mathbf{x}) \in (0, \infty]^{N \times M}$ represent the scores $g(\mathbf{x})$ broadcasted across the base (ACGT) dimension. With these choices of $L(f, \mathbf{X}, P_g)$, conservation penalty, and $P_g(\mathbf{x})$, an inclusion scrambler is trained to output scores in $(0, \infty]^N$ which produce a distribution $P_g(\mathbf{x})$ with maximum entropy but whose samples $\tilde{\mathbf{X}} \sim P_g(\mathbf{x})$ minimize the predictive reconstructive error, $\underset{\tilde{\mathbf{X}} \sim P_g(\mathbf{X})}{\mathbb{E}} \left[ \text{KL} \left[ f(\tilde{\mathbf{X}}) \| f(\mathbf{X}) \right] \right]$, thus identifying sufficient features.

**Occlusion Scrambler.** An *occlusion* scrambler is trained with

$$L(f, \mathbf{X}, P_g) = - \underset{\tilde{\mathbf{X}} \sim P_g(\mathbf{X})}{\mathbb{E}} \left[ \text{KL} \left[ f(\tilde{\mathbf{X}}) \| f(\mathbf{X}) \right] \right] \tag{8}$$

$$\text{Con}(P_g(\mathbf{X}), \tilde{B}, \mathbf{X}) = \left( t_{\text{bits}} - \frac{1}{N} \text{KL} \left[ P_g(\mathbf{X}) \| \mathbf{X} \right] \right)^2 \tag{9}$$

$$P_g(\mathbf{x}) = \sigma(\log(\tilde{B}) + \mathbf{x} / \dot{g}(\mathbf{x})). \tag{10}$$

With these choices of $L(f, \mathbf{X}, P_g)$, conservation penalty, and $P_g(\mathbf{x})$ an occlusion scrambler is trained to output scores in $(0, \infty]^N$ which produce a distribution $P_g(\mathbf{x})$ with minimum entropy but whose samples $\tilde{\mathbf{X}} \sim P_g(\mathbf{x})$ maximize the predictive reconstructive error. Since the samples from $P_g(\mathbf{x})$ maximize the reconstructive error, this formulation identifies necessary features.

## 2.3 SHORTCOMINGS OF SCRAMBLERS

While scramblers outperform common post-hoc methods in providing explanations, they still face some key limitations that affect their overall effectiveness.

**Lack of Key Prior Knowledge.** DNA motifs are generally recognized as small, contiguous, and disjoint subsequences within a larger sequence (Klug, 1995; Alberts et al., 2002; Siggers & Gordân, 2014; Lambert et al., 2018; Stormo, 2013; Maston et al., 2006). Thus, incorporating this key information as an inductive bias into explanation methods could greatly improve the quality of the identified motifs. However, scramblers do not explicitly consider this prior knowledge; instead, they learn a distribution $P_g(\mathbf{X})$ over sequences, which optimizes jointly for the prediction reconstruction error and for entropy. While this formulation is valid to produce necessary or sufficient explanations, it fails to capture prior knowledge of motif biology, particularly that motifs occur as one or more small, contiguous, and disjoint subsequences.

**Limitations of the Conservation Penalty.** While inclusion and occlusion scramblers aim to maximize and minimize the entropy of $P_g(\mathbf{x})$ via their conservation penalty, $\text{CON}(P_g(\mathbf{X}), \tilde{B}, \mathbf{X})$, their effectiveness heavily depends on the choice of the $t_{\text{bits}}$ parameter. This parameter controls the entropy and serves as the target value for the expected entropy of $P_g(\mathbf{X})$. For example, a larger $t_{\text{bits}}$ allows more entropy for $P_g(\mathbf{x})$ with respect to either the background $\tilde{B}$ or the sample $\mathbf{X}$, depending on whether an inclusion or occlusion scrambler is being learned. The challenge with this formulation is that we often do not know how many motifs exist or how distinct they are from the background signal, making it difficult to determine an appropriate target entropy for $P_g(\mathbf{x})$. Additionally, considering our prior knowledge of motif biology, there is no theoretically justifiable reason that enforcing entropy will lead to the identification of small, contiguous motifs, which we will demonstrate in our experimental section.

## 3 MOTIF EXPLAINER MODELS

To address the limitations of scramblers and provide more accurate explanations that highlight contiguous and disjoint motifs, we propose *Motif Explainer Models* (MEMs). This model-based based explanation approach is designed to incorporate the key properties of motifs, better capturing the structure and arrangement of motifs within sequences, and offering a more precise and biologically meaningful interpretation. A MEM is a model $m : \mathcal{X} \mapsto [0,1]^N$ that outputs importance scores in $[0,1]^N$. For a sequence $\mathbf{x}$, a MEM outputs scores $m(\mathbf{x}) = (m_1, \ldots, m_N)$ producing a probability distribution $P_m(\mathbf{x})$. In the formulation of MEMs, $P_m(\mathbf{x})$ is a probability distribution over the random variable $\tilde{\mathbf{X}} = (\tilde{X}_1, \ldots, \tilde{X}_N)$ where

$$\Pr[\tilde{X}_i = x_i] = m_i \quad \text{and} \quad \Pr[\tilde{X}_i = b_i] = 1 - m_i, \tag{11}$$

i.e., $\tilde{X}_i \sim \text{Bernoulli}(m_i)$ with outcomes $\{x_i, b_i\}$. Here $b_i$ are entries of a vector $\mathbf{b} \in \mathcal{X}$, a background vector used to fill the entries of $\tilde{\mathbf{X}}$. A MEM is learned by solving the following general optimization problem

$$\underset{m \subseteq \mathcal{H}}{\arg\min} \quad \underset{\mathbf{X} \sim \mathcal{D}_{\mathcal{X}}}{\mathbb{E}} \left[ L(f, \mathbf{X}, P_g) + R(m(\mathbf{X})) \right] \tag{12}$$

Here, $L(f, \mathbf{X}, P_m)$ is a loss function that measures the reconstruction error between original predictions $f(\mathbf{X})$ and predictions on the samples from $P_m(\mathbf{X})$. The term $R(m(\mathbf{X}))$ is a regularizer that controls the complexity of the MEM outputs. There are two types of MEMs that can be learned: a sufficient MEM (s-MEM) and a necessary MEM (n-MEM), depending on the choice of loss function $L(f, \mathbf{X}, P_M)$. The regularizer $R$ remains the same for both types of MEMs.

**Loss Function.** The choice of loss function determines whether one wants to learn a s-MEM or n-MEM. To learn an s-MEM we utilize the following loss function:

$$L(f, \mathbf{X}, P_m) = \rho\left( f(\mathbf{x}), \mathbb{E}[f(\tilde{\mathbf{X}})] \right). \tag{13}$$

where, $\rho : \mathbb{R} \times \mathbb{R} \mapsto \mathbb{R}$ is a measure of similarity on $\mathbb{R}$ and the expectation is over $P_M$ and $\mathbf{b}$ (if $\mathbf{b}$ is not fixed and instead sampled from some distribution). With this choice of $L$, an s-MEM

minimizes the reconstruction error between the original prediction $f(\mathbf{X})$ and the average prediction over $P_m(\mathbf{X})$. Thus, an s-MEM is specifically designed to identify sufficient sets.

Conversely, a n-MEM is trained using the following loss function:

$$L(f, \mathbf{X}, P_m) = -\rho(f(\mathbf{x}), \mathbb{E}[f(\tilde{\mathbf{X}})]).\tag{14}$$

where the expectation is over $P_{(\mathbf{1}-m(\mathbf{x}))}[1]$ and $\mathbf{b}$. Minimizing this loss is equivalent to maximizing $\rho(f(\mathbf{x}), \mathbb{E}[f(\tilde{\mathbf{X}})])$, implying that an n-MEM is maximizing the reconstruction error between a original predictions $f(\mathbf{X})$ and the average prediction over $P_m(\mathbf{X})$. Therefore, an n-MEM is specifically designed to identify necessary sets.

**Regularizers.** Since motifs are known to be small, contiguous subsequences, we incorporate this key prior knowledge into our MEMs. Unlike scramblers, we regularize our models with an inductive bias that *directly* encourages the identification of disjoint, contiguous regions consisting of a limited number of base pairs.

To construct the regularizer $R$, we draw inspiration from sentiment analysis in natural language processing (Brinner & Zarrieß, 2023). In NLP, disjoint clusters of words typically interact to convey sentiment; similarly, base pairs in DNA sequences interact to form motifs. Indeed, it has been shown that the syntactical structure of genome regulation has many similarities to natural language (Hwang et al., 2024). Following the approach of Brinner & Zarrieß (2023), we assume a linear coordinate system on the input DNA sequences $\mathbf{x}$ and define a distance $d(i, j)$ between base pairs $i$ and $j$. With this assumed structure, instead of having our MEM directly outputting scores $m(\mathbf{x}) = (m_1, \ldots m_N)$, we have it output two vectors $\mathbf{w} \in \mathbb{R}^N$ and $\sigma \in \mathbb{R}^N_{>0}$, and calculate the final scores as follows:

$$m_j = \text{sigmoid}\left(\sum_i w_{i,j}\right) \quad \text{where} \quad w_{i,j} = w_i \cdot \exp\left(-\frac{d(i,j)^2}{\sigma_i}\right)$$

Thus, the optmization is done with respect to $\mathbf{w}$ and $\sigma$. As noted by (Brinner & Zarrieß, 2023), this parameterization of $m(\mathbf{X})$ will encourage neighboring base-pairs to be assigned similar scores if the corresponding $\sigma$ values are large. Large $\sigma$ values promoted with an additional regularization term:

$$\lambda_2 \cdot \frac{1}{N} \sum_i \log(\sigma_i).\tag{15}$$

This allows for sharper boundaries between importance scores for neighboring base-pairs as needed. With this parameterization, the final regularizer $R$ is then defined as

$$R(m(\mathbf{X})) = \lambda_1 \cdot ||m(\mathbf{X})||_1 - \lambda_2 \cdot \frac{1}{N} \sum_i \log(\sigma_i).\tag{16}$$

where $|| \cdot ||_1$ is the $\ell_1$ norm. This regularizer incorporates prior domain knowledge about motifs to enable the MEM to identify them effectively. The first term encourages the importance scores to be sparse, meaning only a small number of base pairs are assigned high scores. The second term encourages neighboring importance scores to be similar while also promoting sharp boundaries when optimal. This is crucial because it allows for the discovery of disjoint contiguous regions, enabling a more accurate representation of the motifs and their distinct properties.

## 4 EXPERIMENTAL RESULTS

We conduct experiments on synthetic DNA sequences $\mathbf{x} \in \{0, 1\}^{500 \times 4}$ containing two motifs, $A$ and $B$, which are the SPI1 and CTCF DNA-binding motifs consisting of 10 and 12 base-pairs, respectively (Friedman, 2007; Pchelintsev et al., 2016). We model three common logical syntax rules—cooperation, repression, and redundancy—to determine the labels $Y \in \{0, 1\}$. We will show that MEMs outperform scramblers in accurately detecting important motifs and deducing the underlying logic.

---

[1]$\mathbf{1}$ is the vector of all 1's in $\mathbb{R}^N$

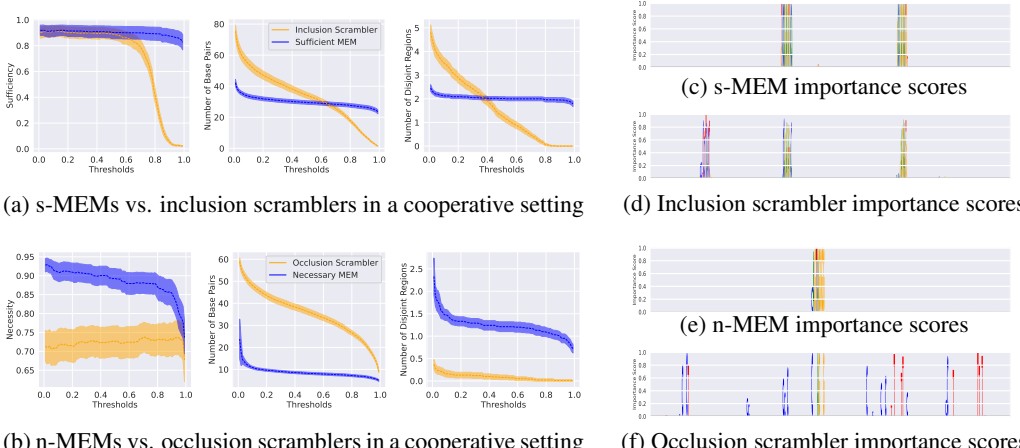

Figure 1: Results on positively labeled sequences ($Y = 1$) under a cooperative syntax

For all three logical rules, we define our base predictor model $f$ (predicting labels from input sequences) to be a residual network with dilated convolutions. To ensure a fair comparison between MEMs and scramblers, we normalize the attribution scores from scramblers to the range $[0, 1]$ by generating PSSMs using Eqs. (7) and (10), computing the information content as per (Shannon, 1948), and then applying min/max normalization. The normalized importance scores yield solution sets $S_t$ by thresholding scores for $t \in (0, 1)$. For any $t$, $S_t$ is represented as a binary vector $s_t \in \{0, 1\}^{500}$, where $(s_t)_j = 1$ if $j \in S_t$ and 0 otherwise.

To compare the effectiveness of MEMs and scramblers, we quantify key properties of the solution sets $S_t$ on an external sample of 100 DNA sequences. We measure the sufficiency and necessity of $S_t$ with $1 - |\hat{Y}(\mathbf{x}) - f_{S_t}(\mathbf{x})|$ and $|\hat{Y}(\mathbf{x}) - f_{\bar{S}_t}(\mathbf{x})|$, where $\hat{Y}(\mathbf{x})$ is the predicted class of model $f$. Higher values of these quantity indicate greater sufficiency and necessity of $S_t$, respectively. Additionally, we count the number of base pairs in $S_t$ as $|S_t| = ||s_t||_0$ and determine the number of disjoint regions by counting clusters of consecutive '1's in $s_t$. A crucial limitation of scramblers is that it is impossible to select an appropriate value of $t$ a priori (the threshold for distinguishing important from non-important features). Thus, we compute our interpretability metrics on scramblers for all $t \in (0, 1)$. As a result, the effectiveness of MEMs and scramblers will be measured by how their ability to generate good explanations over all possible such threshold. We will see that in contrast to MEMs, the performance of scramblers is highly sensitive to the value of $t$, and there is generally no single value of $t$ in any experiment for which scramblers can outperform MEMs. Details on experiment implementation and additional figures are included in Appendices A.1 and A.2

## 4.1 LEARNING LOGICAL SYNTAX

We consider the three following types of logical syntax between motifs. These three arguably constitute the vast majority of syntactical constraints between motifs in regulatory biology.

| **Cooperative** | **Redundant** | **Repressive** |
|---|---|---|
| $Y = \begin{cases} 1 & \text{if } A \wedge B \\ 0 & \text{otherwise} \end{cases}$ | $Y = \begin{cases} 1 & \text{if } A \vee B \\ 0 & \text{otherwise} \end{cases}$ | $Y = \begin{cases} 1 & \text{if } M_1 \wedge \neg M_2 \\ 0 & \text{otherwise} \end{cases}$ |

### 4.1.1 COOPERATIVE SYNTAX

We begin by considering a data-generating process that follows a cooperative syntax. This rule assigns a positive label only when both motifs $A$ and $B$ are present, resulting in a negative label otherwise. Consequently, for a predictor trained on this rule, the set $\{A, B\}$ should be deemed sufficient for positive predictions $\hat{Y} = 1$, as the true data-generating process necessitates the presence of both motifs to produce a positive label. Conversely, amongst the positive predictions, either set, $\{A\}$ or $\{B\}$, is necessary because without both, the true data-generating process produces a negative label.

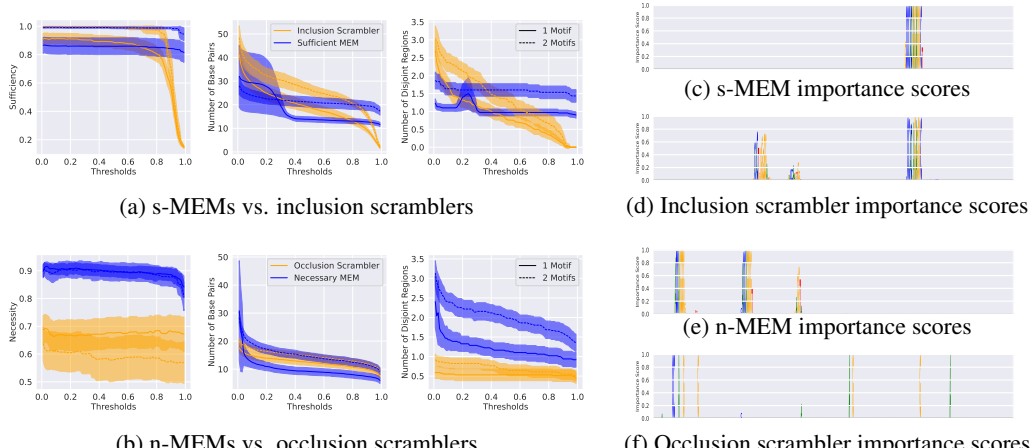

Figure 2: Results on positively labeled sequences ($Y = 1$) under a redundant syntax

In Fig. 1a, we compare the effectiveness of s-MEMs and inclusion scramblers in explaining the predictor and recovering the correct set of sufficient motifs. For the sequences with true label $Y = 1$, the results show that for thresholds $t \in (0, 0.6)$, both methods successfully identify sufficient features. However, as $t$ increases, the inclusion scrambler struggles to recover the sufficient set. Notably, the s-MEM is more accurate and outperforms the inclusion scrambler because it identifies two motifs as being sufficient for the predictor. Across all $t \in (0, 1)$, the s-MEM identifies approximately 20–30 important base pairs across 2–3 disjoint regions, while the inclusion scrambler detects between 0 and 80 base pairs, with 0–6 regions depending on the threshold $t$.

In Fig. 1b, we compare the effectiveness of n-MEMs and occlusion scramblers in recovering the correct necessary motifs. The results indicate that, for all thresholds both methods identify necessary regions, with the n-MEM detecting more necessary regions. More importantly though, the n-MEM is identifying motifs while the scrambler is not. The n-MEM is identifying 0-20 important base-pairs dispered in 1-1.5 regions, while the scrambler is detecting anywhere from 0-60 base-pairs dispersed randomly, as indicated by identify a 0-0.5 regions on average. Examples are provided in Figs. 1e and 1f.

By combining the interpretations from our s-MEM and n-MEM, we are able to accurately and robustly identify the that 2 motifs are sufficient and 1 is necessary for a positive prediction. Thus, we can deduce that this setting indeed follows a cooperative syntax.

### 4.1.2 REDUNDANT SYNTAX

We next consider a redundant syntax setting. This rule assigns a positive label if either $A$, $B$, or both are present, and a negative label otherwise. As a result, for a predictor that effectively learns this rule, either the sets $\{A\}$ and $\{B\}$ are sufficient since the true data-generating process assigns a positive label when either motif is present. On the other hand, depending on whether a sequence contains either $A$ or $B$ or both, the set of necessary motifs may vary. For sequences that contain both, the set $\{A, B\}$ is necessary since only when both are removed will the predictor generate predictions that yield a classification $= 0$. For a sequence that contains only $A$ (or $B$), the set $\{A\}$ (or $\{B\}$) is necessary as the removal of this single motif will render the label $Y = 0$.

In Fig. 2b, we compare s-MEMs and inclusion scramblers in a redundant setting. The results show that for thresholds $t \in (0, 0.9)$, both methods identify sufficient regions; however, as $t$ increases, the inclusion scrambler struggles to recover sufficient regions. Notably, for sequences labeled $Y = 1$ due to a single motif (either $A$ or $B$), both methods identify sets that are slightly less sufficient compared to those for positively labeled sequences containing both motifs. More importantly though, for the $Y = 1$ sequences with a single motif, the s-MEM is able to detect 1 disjoint region for nearly all $t$ while the inclusion scrambler identifies more regions for smaller $t$ and less regions for large $t$. Likewise, for sequences with a ground truth of two motifs, the sufficient MEM detects 20-30 base-pairs that are dispersed in 1.5 to 2 regions. Note, theoretically, one motif is sufficient to predict

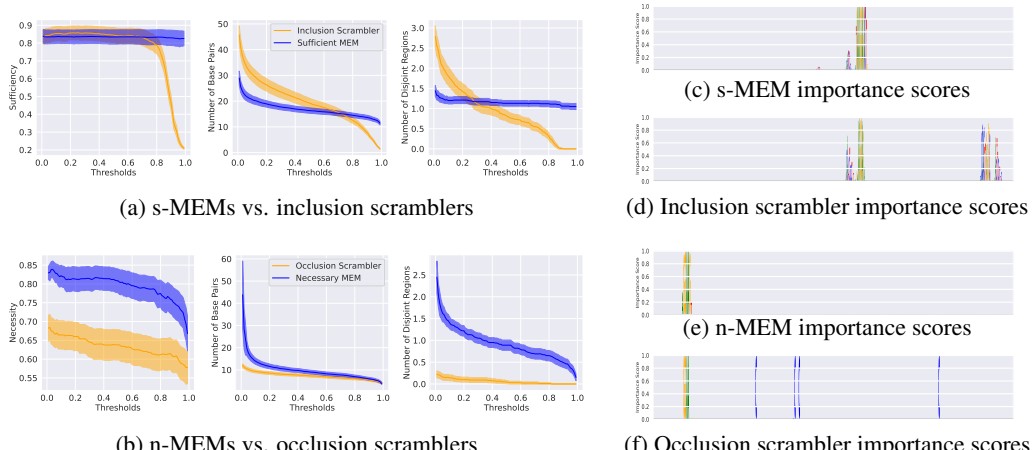

(a) s-MEMs vs. inclusion scramblers

(c) s-MEM importance scores

(d) Inclusion scrambler importance scores

(b) n-MEMs vs. occlusion scramblers

(e) n-MEM importance scores

(f) Occlusion scrambler importance scores

Figure 3: Results on positively labeled sequences ($Y = 1$) under a repressive syntax

the positive label but the s-MEM identifies a bit more. We attribute this to the s-MEM's learning that there are two motifs present in this sequence and it attributing some importance to the second motif.

In Fig. 2e, we highlight how n-MEMs are able to identify necessary motifs with much greater success than occlusion scramblers. The results show that across all thresholds, both methods identify necessary regions, with n-MEMs identifying regions that are much more necessary. Additionally, n-MEMs are able to accurately detect the correct the number of motifs amongst the two modes of the ground truth (i.e., whether there is 1 or 2 motifs). As expected, for sequences with two motifs, our n-MEM identifies 10-30 base-pairs and 2-2.5 regions for many $t$, while for sequences with only one motif, the n-MEM identifies 5-10 base-pairs and that make up 1-1.5 regions to be necessary. On the other hand, the occlusion scrambler fails to distinguish these details. Instead, it outputs the same (incorrect) explanations for both modes of the ground truth, identifying 20-30 base pairs making up 0.5-1 regions to be necessary.

Thus, by using a s-MEM and n-MEM, we are able to accurately identify the that 1 motif is sufficient and 1-2 motifs is necessary (depending on the sequence) for a positive prediction. Therefore, we can conclude this setting indeed follows a redundant syntax.

### 4.1.3 REPRESSIVE SYNTAX

Lastly, we consider a data-generating process based on a repressive syntax. This rule assigns a positive label if $M_1$ is present and $M_2$ is absent, and a negative label in all other cases. The logic in this rule is more involved as $M_2$ represses $M_1$ from generating a positive labeling. In this setting, for sequences with $Y = 1$, the smallest sufficient and necessary set is $\{M_1\}$ since its sole presence results in a positive classification and removal in a negative classification. On the other hand, for the subset of negatively labeled sequences, $Y = 0$, the logic is more involved. When the label $= 0$ due to both $M_1$ and $M_2$ being present, the sufficient and necessary set is $M_2$ because its presence yields the correct negative prediction and its removal results in the sequence having only $M_1$ present which yields a positive label. For the subset of negatively labeled sequences that contains $M_2$ only, the set $\{M_2\}$ is both sufficient and necessary.

In Figs. 3a and 3b, we compare the ability of MEMs and scramblers to identify the sufficient and necessary motifs on sequences with $Y = 1$. In, Fig. 3a we see for thresholds $t \in (0, 0.8)$, both methods identify sufficient regions but as $t$ continues to increase, the inclusion scrambler fails to recover sufficient regions. Furthermore, the s-MEM outperforms the scrambler for $t \in (0, 0.8)$ in correctly identifying a single motif. The s-MEM identifies 15-30 important base pairs dispersed in 1-1.5 regions while the scrambler inaccurately dentifies 15-50 important base pairs dispersed anywhere from 0.5-3 regions. In Fig. 3b, both n-MEMs and occlusion scramblers identify necessary base-pairs with the n-MEM identifying those that are more necessary. Interestingly, the occlusion scramblers identify a smaller number of important base-pairs. However, for $t \in [0.1, 0.9]$ the n-MEM detects

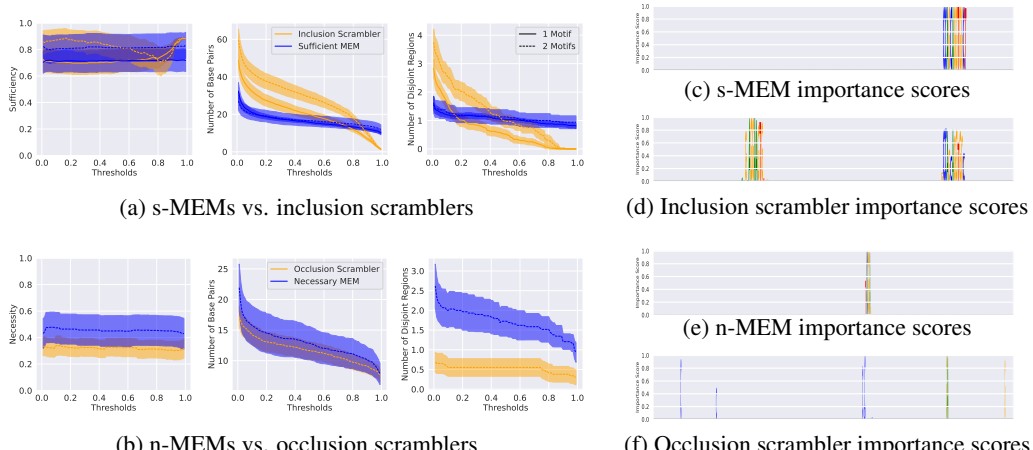

Figure 4: Results on negatively labeled sequences ($Y = 0$) under a repressive syntax

0.5-1.5 regions while the occlusion detects nearly no regions on average. This suggests that the occlusion scrambler is erroneously identifying random base-pairs as necessary and not the actual important motifs. One can see an example of this in Fig. 3f

In Fig. 4b, we highlight how n-MEMs are able to indeed identify necessary motifs for the subpopulation of sequences that have label $Y = 0$ due to the presence of both $A$ and $B$. The results show that both methods identify necessary regions, with necessary explainers identifying regions that are more necessary. Additionally, both methods identify roughly the same number of important base-pairs, which ranges form 5-25. However, the n-MEM is able to discern that there are 1-2 important regions (i.e. the B motif) while the occlusion scrambler cannot, as noticed by it identifying 0-1 important regions. An example of this is illustrated in Fig. 4f

In conclusion, by using an s-MEM and n-MEM, we are able to accurately discern that, for positive predictions, one motif ($A$) is both sufficient and necessary. Additionally, for negative predictions, there exists a sub population of sequences that for which one motif, $B$, is sufficient. Furthermore, amongst this sub-population there exists sequences for which 1 motif, $B$, is necessary where removing it generates a positive prediction, (implying $A$ was repressed by $B$). Thus, we can ultimately deduce that this setting indeed follows a repression syntax.

## 5 CONCLUSION & FUTURE DIRECTIONS

In this work, we introduced Motif Explainer Models (MEMs), a novel explanation method for genomic DNNs that identifies both sufficient and necessary motifs in complex DNA sequences. In contrast to current methods like scramblers, MEMs leverage prior domain knowledge as an inductive bias to cleanly identify individual motifs as disjoint and contiguous subsequences. Furthermore, by discovering sufficient and necessary motifs separately, MEMs address the limitations of existing post-hoc methods that often fail to capture the intricate logical relationships between motifs. Our approach not only improves the interpretability of genomic DNNs, but also uncovers the logical syntax governing gene regulation, distinguishing between as cooperative, repressive, and redundant interactions.

Through extensive experiments, we demonstrated that MEMs outperform current methods in detecting important motifs and deciphering their underlying syntax. By providing more accurate and comprehensive explanations, MEMs offer new insights into the functional roles of motifs in gene regulation, paving the way for better understanding of transcription-factor binding and genomic activity. In summary, MEMs represent a significant step forward in interpreting complex genomic models, offering a robust framework for elucidating the logic behind motif interactions. Future work may explore extending this framework to more diverse regulatory contexts, ultimately enhancing our ability to interpret the functional landscape of the genome.

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

# A   APPENDIX

## A.1   ADDITIONAL EXPERIMENTAL DETAILS

**Implementation of MEMs.**   To learn s-MEMs or n-MEMs $S$ we solve the following optimzation problem

$$\arg\min_{m \subseteq \mathcal{H}} \quad \mathbb{E}_{\mathbf{X} \sim \mathcal{D}_{\mathcal{X}}} \left[ L(f, \mathbf{X}, P_g) + \lambda \cdot R(m(\mathbf{X})) \right] \tag{17}$$

Where to learn s-MEMS we let

$$L(f, \mathbf{X}, P_m) = |\hat{Y}(\mathbf{X}) - \mathbb{E}_{P_m}[f(\tilde{\mathbf{X}})]| \tag{18}$$

and to learn n-MEMS we let

$$L(f, \mathbf{X}, P_m) = -|\hat{Y}(\mathbf{X}) - \mathbb{E}_{P_{1-m}}[f(\tilde{\mathbf{X}})]. \tag{19}$$

We solve this problem via empirical risk minimization. Given $N$ samples $\{\mathbf{X}_i\}_{i=1}^{N} \overset{\text{i.i.d.}}{\sim} \mathcal{D}_X$, we learn a model $m$ to minimize

$$\frac{1}{N} \sum_{i=1}^{N} \left[ L(f, \mathbf{X}_i, P_m) + \lambda \cdot R(m(\mathbf{X}_i)) \right] \tag{20}$$

where

$$\mathbb{E}[f(\tilde{\mathbf{X}}_\mathbf{i})] = \frac{1}{K} \sum_{j=1}^{K} f((\tilde{\mathbf{X}}_i)_j). \tag{21}$$

In theory, the entries of $(\tilde{\mathbf{X}}_i)_j$ are Bernoulli($m_i$) with outcomes $\{x_i, b_i\}$. where $b_i$ are entries of a vector $\mathbf{b} \in \mathcal{X}$, a background vector used to fill the entries of $\tilde{\mathbf{X}}$. In practice, to allow for differentiaion during optimization, we generate discrete samples using the Gumbel-Softmax distribution. During optimization we set $K = 10$.

Recall the form of regularizer

$$R(m(\mathbf{X})) = \lambda_1 \cdot ||m(\mathbf{X})||_1 - \lambda_2 \cdot \frac{1}{N} \sum_{i} \log(\sigma_i). \tag{22}$$

To learn MEMs use a residual network with dilated convolutions. To learn s-MEMs, we set $\lambda_1 = 2$ and $\lambda_2 = 0.5$. To learn n-MEMs, we set $\lambda_1 = 5$ and $\lambda_2 = 0.01$. We used a batch size of 32 and trained for each MEM for 25 epochs using an Adam optimizer with default $\beta$-parameters of $\beta_1 = 0.9$, $\beta_2 = 0.99$ and a fixed learning rate of 0.001.

**Implementation of Scramblers.** To learn inclusion and occlusion scramblers we simply follow the protocol in Linder et al. (2022) and use a residual network with dilated convolutions. To learn inclusion scramblers, we set $\lambda = 2$ and $t_{\text{bits}} = 1 \times 10^{-4}$. To learn occlusion scramblers, we set $\lambda = 5$ and $t_{\text{bits}} = 1 \times 10^{-4}$. We use a batch size of 32 and train for 25 epochs using an Adam optimizer with default $\beta$-parameters of $\beta_1 = 0.9$, $\beta_2 = 0.99$ and a fixed learning rate of 0.001.

## A.2 ADDITIONAL FIGURES

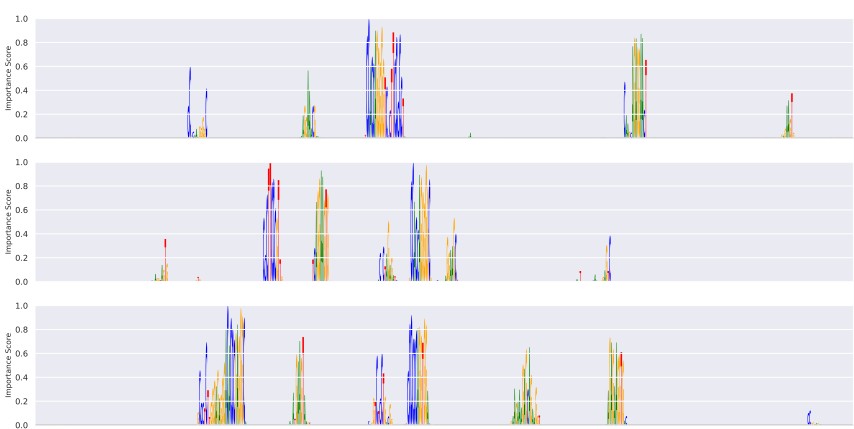

Figure 5: Inclusion scrambler importance scores for a cooperative syntax

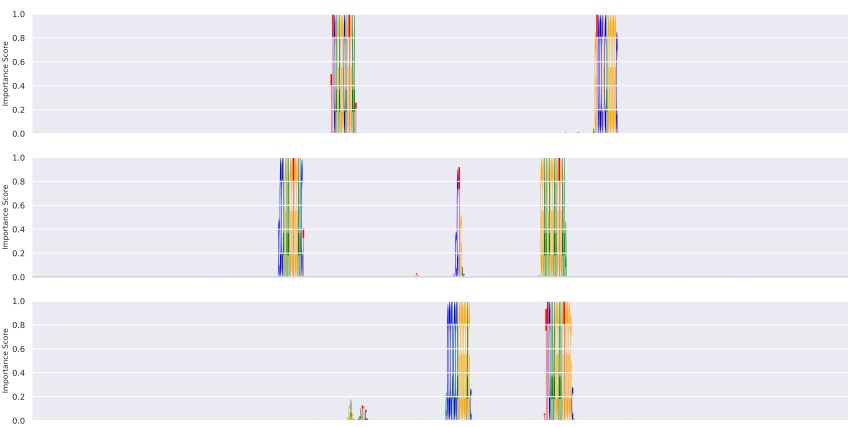

Figure 6: s-MEM importance scores for a cooperative syntax

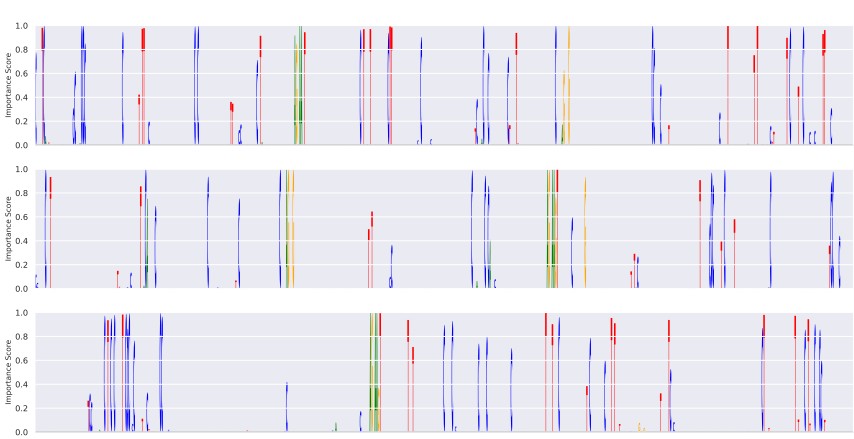

Figure 7: Occlusion scrambler importance scores for a cooperative syntax

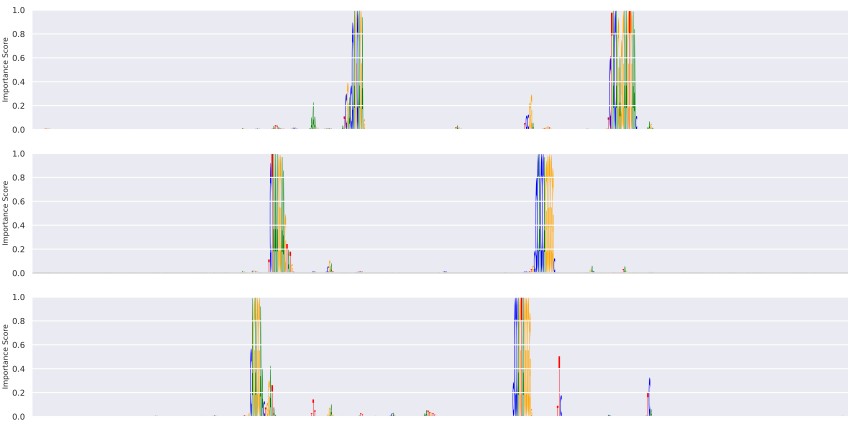

Figure 8: n-MEM importance scores for a cooperative syntax

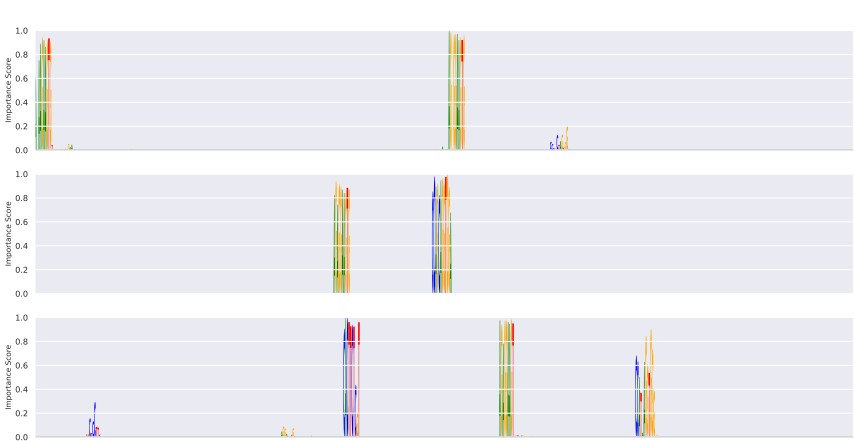

Figure 9: Inclusion scrambler importance scores for a redundant syntax

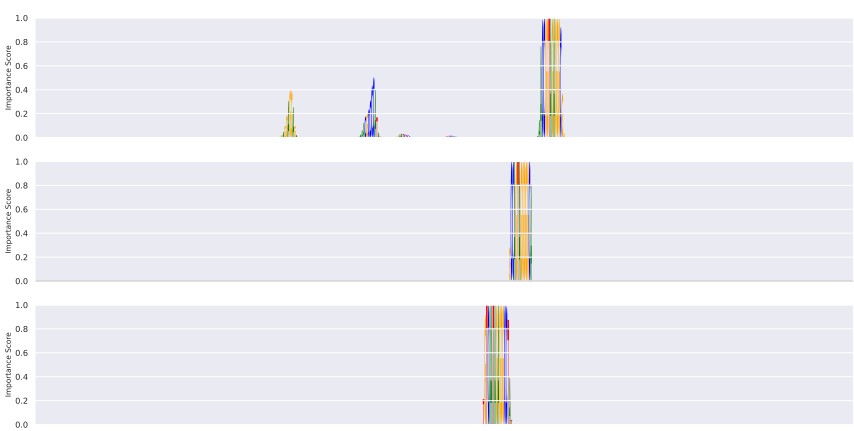

Figure 10: s-MEM importance scores for a redundant syntax

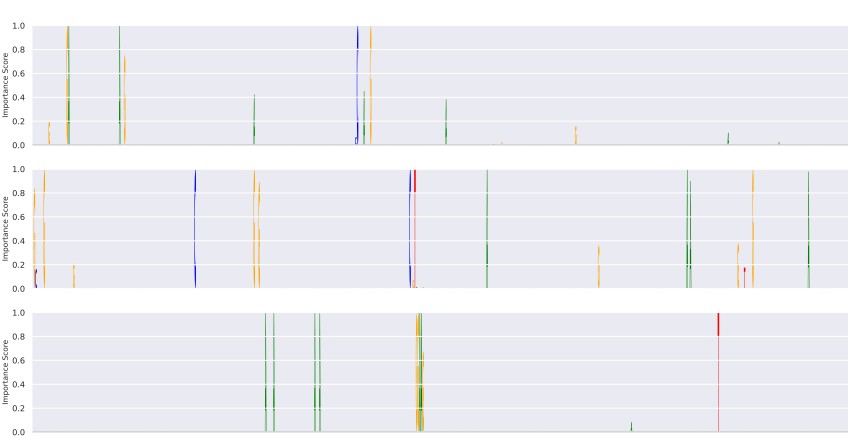

Figure 11: Occlusion scrambler importance scores for a redundant syntax

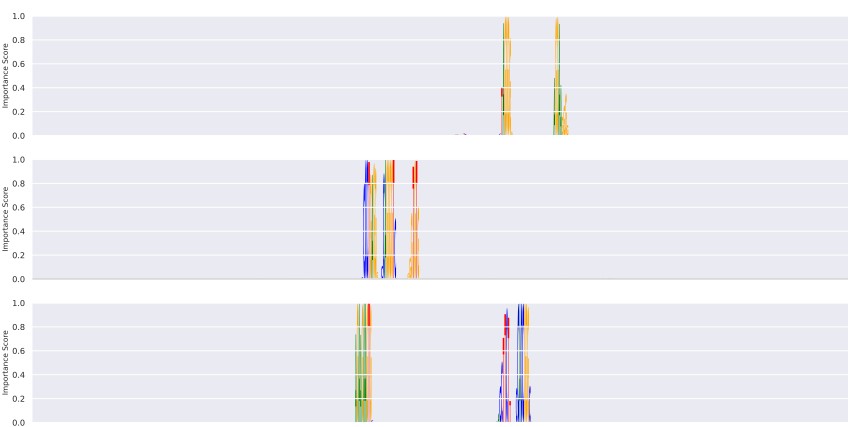

Figure 12: n-MEM importance scores for a redundant syntax

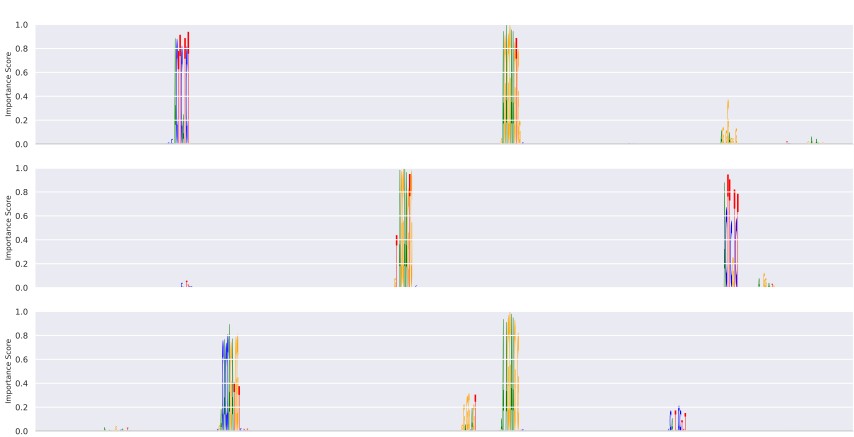

Figure 13: Inclusion scrambler importance scores for a repressive syntax

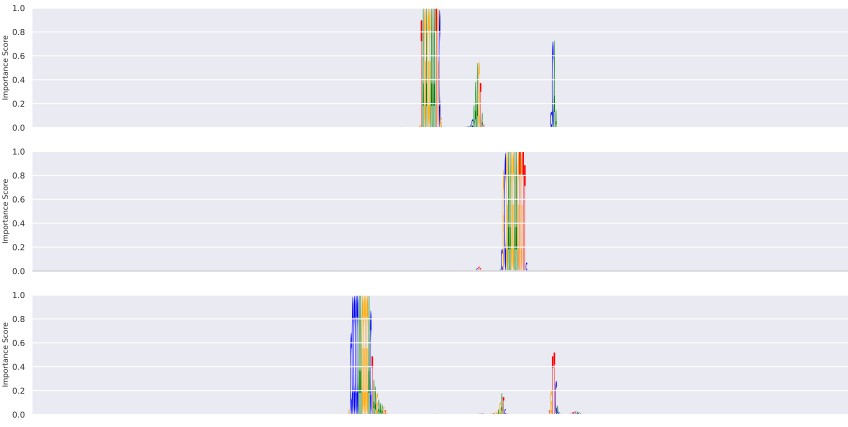

Figure 14: s-MEM importance scores for a repressive syntax

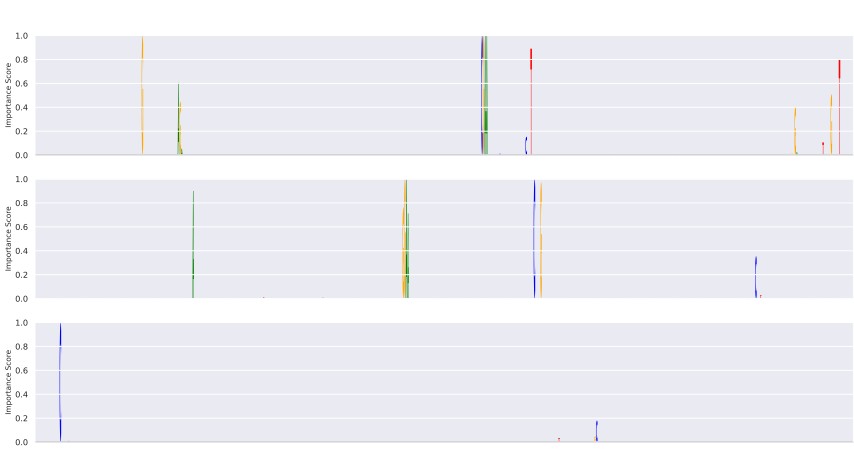

Figure 15: Occlusion scrambler importance scores for a repressive syntax

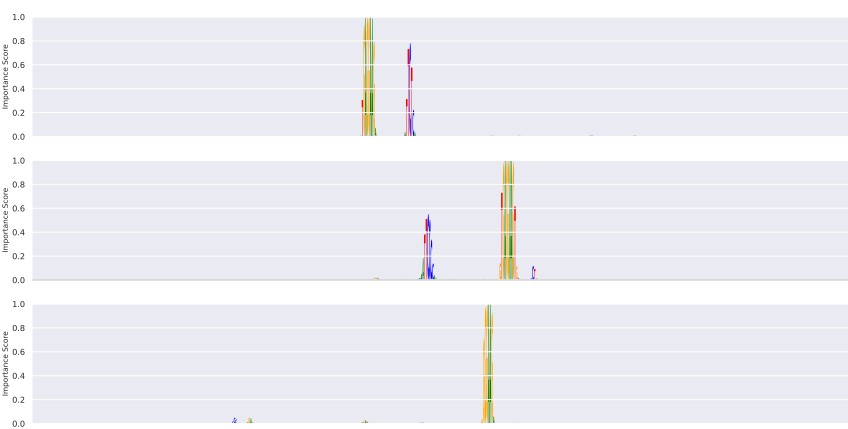

Figure 16: n-MEM importance scores for a redundant syntax

