# OpenReview forum: "Uncovering BioLOGICAL Motifs and Syntax via Sufficient and Necessary Explanations"
_ICLR.cc/2025/Conference — Submitted to ICLR 2025_

### Official Review · Reviewer_NtwQ · 2024-10-28

**Soundness:** 1
**Presentation:** 2
**Contribution:** 1
**Rating:** 1
**Confidence:** 5

**Summary:**

The authors consider the problem of understanding the transcription factor—DNA binding code in terms of binding motifs in the genome. Defining a framework for a logical motif syntax, they aim to find necessary and sufficient motif rules that explain a predictor’s decision making in interpretable. On synthetic data, they evaluate their method against an existing approach.

**Strengths:**

- The methodological part is well written and easy to follow.

**Weaknesses:**

- This paper excessively uses definitive statements that often turn out false. Already in the abstract, to motivate their work they state “TF binding is driven by sequence motifs, […]”, which is not true. A large part of TF binding is driven by other proteins, including pioneering and recruiting factors, but more importantly by the chromatin state – if the chromatin is closed, most Tfs can’t bind even though a motif is present in the sequence. Efforts such as the large-scale consortium-based benchmarking in ENCODE-DREAM challenge [1], have shown that Epigenetics (e.g., openness of chromatin) play a decisive role in the binding of transcription factors and need to be taken into account for accurate prediction [2], a versatile set of tools has hence been designed including [3] to appropriately consider such data. Similarly, in the paragraph on “Lack of Key prior knowledge” that motivates the design choice of their framework, they state that DNA motifs are generally recognized as small, contiguous, and disjoint subsequence […]”, which is also not true. Tfs that have similar roles or are in competition often share parts of their motifs, hence the subsequences are not disjoint. Moreover, important factors such as REST, do have several parts that separately bind to DNA (left and right half of motif), hence the motif is not necessarily continuous. This means the designed regularizers do not make sense.

- Apart from scramblers, the paper lacks references to important work in the scope of XAI. Standard methods for CNNs such as layer-wise-relevance propagation or gradient-based attribution methods (SmoothGrad etc) could equally be used to extract motifs from a pre-trained model, which is neither referenced, nor discussed, or compared to. Contrary to what the authors write here, they do not consider features (here base pairs) in isolation and they *are* able to discover the complex interactions between base pairs and motifs. Other methods, such as inherently interpretable NNs for the considered problem specifically [4], are also neither mentioned nor compared to.

- It is unclear why motifs need to be learned, as there exist extensive motif databases (TRANSFAC, JASPAR, ...) that were generated through rigorous (wet-lab) experiments. Why is it better to use an (inaccurate) inferred motif than detecting motif occurrence and then building a classifier based on that? The paper would benefit from a use-case where the existing databases can not help (e.g., TFs where motifs are not available, or relevant species were no database is available).

- The evaluation is limited – it neither considers relevant methods (see above) nor a real dataset. It also does not consider a downstream application. Each of these would be a necessary condition for a good paper.



[1] https://www.synapse.org/Synapse:syn6131484/wiki/402037, and associated conference tracks at RECOMB/ISCB 2016

[2] J Keilwagen, S Posch, J Grau, Accurate prediction of cell type-specific transcription factor binding Genome Biology 2019

[3] F Schmidt, F Kern, MH Schulz, Integrative prediction of gene expression with chromatin accessibility and conformation data Epigenetics & Chromatin 2020

[4] AM Tseng et al., A MECHANISTICALLY INTERPRETABLE NEURAL NET-
WORK FOR REGULATORY GENOMICS arXiv:2410.06211v 2024

**Questions:**

- See weaknesses above, and

- How well do you recover an *actual* ground truth motif? It seems your main evaluation is necessity and sufficiency with respect to the classifier, but how well are the motifs recovered?

- How robust is your method to noise? How good on real data?

**Details Of Ethics Concerns:**

Large parts of the related work section are almost identical to [1]. Only minimal efforts have been made to change the content (change order of citation, change of individual words, adding a paragraph title in front of a section)

[1] AM Tseng et al., A MECHANISTICALLY INTERPRETABLE NEURAL NET-
WORK FOR REGULATORY GENOMICS arXiv:2410.06211v 2024

---

### Official Review · Reviewer_xokd · 2024-11-02

**Soundness:** 2
**Presentation:** 2
**Contribution:** 2
**Rating:** 3
**Confidence:** 3

**Summary:**

This paper proposes a model-based explanation method — Motif Explainer Models — to identify important motifs and their syntax from deep neural networks trained on genomic data. The method can distinguish between sufficient and necessary motifs, and improves on prior work by better modeling sequences with multiple motifs

**Strengths:**

- This paper tackles an important problem for genomic deep neural networks - explaining their predictions to understand biological motif syntax.
- The approach presented in the paper is well biologically motivated, and attempts to separately model different aspects of motif syntax, such as cooperativity, redundancy, and repression, and employs novel regularization to directly encourage the explainer model to identify disjoint, contiguous motifs.

**Weaknesses:**

- Overall the approach is interesting, but lacks novelty as it is very similar to the scrambler approach with a different loss and penalty. In addition, all of the results use an entirely synthetic dataset of very contrived simple settings. While these are informative to show as a proof of concept, additional results on real biological datasets would make the argument that the method is useful, and a significant advance over scramblers, much more compelling.
- Many of the descriptions lack clarity and are hard to follow, I’ve highlighted a few in the questions section below.

**Questions:**

- In equation 1, why does f() on the righthand side of the equation take two inputs? Are the two vectors disjoint subsets of the N positions?
- A large amount of the paper is spent re-explaining the scramblers approach from Linder et al. (2022) - is this necessary?
- In all of the figures, the sequence logo plots are very difficult to see
- for the y=0 repressive case, explanation of the results doesn’t match up with theoretical explanation of which motifs should be necessary and sufficient. When the experiment is presented, on line 474 it says that only M2 should be sufficient and necessary if M1 and M2 are present, but in the results, line 511-512 it says that the n-MEM is able to discern that 1-2 motifs are necessary.
- In Figure 4b, the 1 motif case seems to be missing

Minor:
- L116: The term “Experimental validation” is misleading because it sounds like there might be biological experiments to validate the model's predictions. Perhaps “Simulation experiments” is more clear?
- L244 has a typo. There is an extra word “based”
- Not all equations in the text are numbered
- For the equation between 14 and 15 (missing a number) - the function d does not seem to be defined anywhere in the text

---

### Official Review · Reviewer_Stmx · 2024-11-03

**Soundness:** 2
**Presentation:** 2
**Contribution:** 2
**Rating:** 5
**Confidence:** 4

**Summary:**

This study proposes a novel model-based explanation method called Motif Explainer Models (MEMs), aimed at interpreting genomic deep neural networks (DNNs) to identify essential motifs in DNA sequences and their logical syntax. MEMs utilize sufficiency and necessity criteria to pinpoint crucial motifs and the interactions among them, such as cooperative and repressive interactions. The study evaluates MEMs against state-of-the-art scrambler models across various experimental scenarios to demonstrate superior performance in detecting motif syntax and logical relationships in complex regulatory settings.

**Strengths:**

- The study introduces a new application of sufficient and necessary conditions within DNN-based models to interpret genomic data. By addressing motif syntax, MEMs could potentially advance explainable AI (XAI) within genomics, allowing for novel insights into DNA motif interactions.
- Comprehensive experimental setups, including synthetic DNA sequences and logical syntax rules, provide a good base for comparing MEMs with existing scrambler models. The authors use multiple metrics to measure model performance in detecting motifs and their syntactical patterns, such as sufficiency and necessity scores.
- If successful in broader applications, MEMs could have a notable impact on regulatory genomics by facilitating more interpretable models for motif discovery and understanding genomic regulation patterns.

**Weaknesses:**

- While MEMs are tested extensively on synthetic data with predefined logical rules, the study lacks real biological data application.
- Although the authors test MEMs on logical rules like cooperation and repression, these predefined rules might oversimplify the complexity of motif interactions.
- MEMs rely on intricate mathematical definitions of sufficiency and necessity, which may limit their interpretability for biologists who are not deeply versed in machine learning.
- MEMs’ effectiveness depends on careful selection of hyperparameters, which could affect their scalability to large genomic datasets.

**Questions:**

- The study heavily relies on synthetic sequences and motifs with fixed logical rules, which may not capture the variability seen in real genomic sequences. Applying MEMs to public biological datasets (e.g., ENCODE) could provide a better understanding of their practical utility.
- Could MEMs be extended to capture more complex interactions among motifs?
- MEMs require careful tuning of hyperparameters for each experimental setup. For scalability, it is essential to understand how these parameters generalize, especially in high-dimensional, large-scale datasets typical in genomic studies.
- Including domain-specific priors, such as motif conservation scores, could enhance MEMs’ interpretability. Has the team considered incorporating such biological constraints?

---

### Meta-Review · Area_Chair_WgEV · 2024-12-18

**Metareview:**

This paper presents a model-based explanation method, named MEMs, to identify important motifs and their syntax from deep neural networks trained on genomic data. Experimental results show that MEMs is able to identify multiple disjoint motifs across DNA sequences, which overcomes some limitations of existing methods. Reviewers agreed that this paper studies an important problem. However, reviewers also raised many concerns regarding technical contributions, paper writing, related work, experiments, etc. For instance, the proposed method is similar to the scrambler approach with a different loss and penalty, the evaluation part of this paper lacks real biological data application, and some important references are missing. The authors did not submit rebuttal to address these questions.

**Additional Comments On Reviewer Discussion:**

The authors did not provide responses to address questions from reviewers.

---

### Decision · Program_Chairs · 2025-01-22

Reject